# Robust fractional quantum Hall effect in the $N = 2$ Landau level in bilayer graphene

Georgi Diankov[1,*], Chi-Te Liang[1,2,*], François Amet[3,4], Patrick Gallagher[1], Menyoung Lee[1], Andrew J. Bestwick[1], Kevin Tharratt[1], William Coniglio[5], Jan Jaroszynski[5], Kenji Watanabe[6], Takashi Taniguchi[6] & David Goldhaber-Gordon[1]

The fractional quantum Hall effect is a canonical example of electron–electron interactions producing new ground states in many-body systems. Most fractional quantum Hall studies have focussed on the lowest Landau level, whose fractional states are successfully explained by the composite fermion model. In the widely studied GaAs-based system, the composite fermion picture is thought to become unstable for the $N \geq 2$ Landau level, where competing many-body phases have been observed. Here we report magneto-resistance measurements of fractional quantum Hall states in the $N = 2$ Landau level (filling factors $4 < |\nu| < 8$) in bilayer graphene. In contrast with recent observations of particle–hole asymmetry in the $N = 0/N = 1$ Landau levels of bilayer graphene, the fractional quantum Hall states we observe in the $N = 2$ Landau level obey particle–hole symmetry within the fully symmetry-broken Landau level. Possible alternative ground states other than the composite fermions are discussed.

[1] Department of Physics, Stanford University, Stanford, California 94305, USA. [2] Department of Physics, National Taiwan University, Taipei 106, Taiwan. [3] Department of Physics, Duke University, Durham, North Carolina 27708, USA. [4] Department of Physics and Astronomy, Appalachian State University, Boone, North Carolina 28608, USA. [5] National High Magnetic Field Laboratory, Tallahassee, Florida 32310, USA. [6] Advanced Materials Laboratory, National Institute for Materials Science, 1-1 Namiki, Tsukuba 305, Japan. * These authors contributed equally to this work. Correspondence and requests for materials should be addressed to C.-T.L. (email: ctliang@phys.ntu.edu.tw) or to D.G.-G. (email: goldhaber-gordon@stanford.edu).

A fractional quantum Hall (FQH) state was first observed at Landau level (LL) filling factor $v = 1/3$ in a GaAs/AlGaAs two-dimensional electron system[1]. This many-body state was successfully explained by the Laughlin wave-function[2]. Within the lowest LL, the $v = 2/3 = 1 - 1/3$ state was interpreted as the particle–hole conjugate of the $v = 1/3$ Laughlin state[3]. In the lowest LL, the observed fractional states at filling factors $v = p/(2mp \pm 1)$, with $m$ and $p$ positive integers, can be successfully explained by the composite fermion (CF) picture[4], in which an even number of magnetic flux quanta are attached to an electron. In addition to the CF model, a hierarchical scheme of parent and daughter states[5,6] and a model based on cyclotron braid subgroups[7] offer competing explanations for the pronounced FQH states at $v = p/(2mp \pm 1)$ in the lowest LL.

Graphene, whose band structure leads to the manifestation of relativistic quantum mechanical effects in the solid state[8–10], has also revealed a rich FQH effect[11,12], in which combined spin and valley degrees of freedom lead to new ground states, including multicomponent FQH states[13] with an unconventional sequence[14]. Numerous FQH states have been observed in the $N = 0$ and $N = 1$ LLs of monolayer graphene[13,15] but not in the $N = 2$ LL ($6 < |v| < 10$ for monolayers).

With advances in sample preparation, the FQH effect was also recently seen in bilayer graphene, revealing surprising results such as tunability of states with electric field normal to the plane[16], indications of even-denominator FQH states[17] at $v = -1/2$ and at $v = -5/2$ and, in scanning compressibility measurements, particle–hole asymmetry in the $N = 0/N = 1$ LLs and incipient FQH states in the $N = 2$ LL at $v = 14/3, 17/3, 20/3$ and $23/3$, not forming a complete CF sequence[18]. Highlighting the role of sample-to-sample variability, both symmetric and asymmetric FQH states were seen in the same measurement set in another study[19].

Here we report observations of particle–hole symmetric FQH states in the $N = 2$ LL in bilayer graphene. In contrast, in high ($N \geq 2, v > 4$) LLs of GaAs-based two-dimensional electron systems, aside from possible evidence for $v = 4 + 1/5$ and $v = 4 + 4/5$ FQH states[20], competing charge-ordered states such as Wigner crystal bubbles and nematic stripes are thought to be the many-body ground states[21–23]. Why might charge-ordered states be expected to supplant FQH and specifically CF states in high LLs? In high ($N > 0$) LLs, the more extended electron wave-functions may destabilize the FQH states[21–25]. In GaAs, such wave-functions have nodes at particular momenta corresponding to spatial separation between orbitals on the order of the magnetic length, favouring charge ordering with that spacing[23]. In bilayer graphene, the wave-functions in the $N = 2$ LL have no complete nodes and hence might be expected to support FQH states over charge-ordered states.[26] A numerical study that does not rely on the mean-field approximation or otherwise assume the CF picture predicts pronounced single-component FQH states at 1/3, 2/3 and 2/5 in the $N = 2$ LL[26].

## Results

**Sample characterization.** Observation of the FQH effect requires ultra-clean systems with disorder energy scale smaller than the energy gaps of the elementary excitations from the fractional ground states. We achieve the desired cleanliness by fabricating open-face bilayer graphene/hexagonal boron nitride (h-BN)/graphite stacks sitting on $SiO_2$ (Fig. 1a and Supplementary Fig. 1), specifically avoiding encapsulation of the bilayer graphene with a top h-BN layer in order to keep the dielectric constant low and thus enhance Coulomb interactions (Supplementary Note 1 and Supplementary Fig. 2). All

devices studied in this work were operated at densities $\sim 1.5$–$5 \times 10^{12} \, cm^{-2}$ with zero-field mobility of 100,000–250,000 $cm^2 \, V^{-1} \, s^{-1}$. In zero-field measurements, we typically observe (Fig. 1b) strongly insulating behaviour near the charge neutrality point as previously seen[19,27], with a width of the charge neutrality (Dirac) peak suggestive of low ($\sim 10^{10} \, cm^{-2}$) disorder density[15]. The longitudinal ($R_{xx}$) and Hall ($R_{xy}$) magnetoresistances at a constant density on the hole side of the $N = 2$ LL of device 1 are shown as a function of magnetic field from 11.4 to 45 T at $T = 0.4$ K (Fig. 1c). Confirming the low disorder in the sample, the onset of broken symmetry occurs by 2 T (Fig. 1c, inset) and fully symmetry-broken Hall plateaux are seen by 5 T (Supplementary Note 2 and Supplementary Fig. 3).

**FQH effect.** We observe the FQH effect in the $N = 2$ LL for $4 < |v| < 6$ in pronounced $R_{xx}$ minima at fractional filling $v = -13/3, -14/3, -16/3$ and $-17/3$, with accompanying plateau-like structures in $R_{xy}$ at $v = -13/3$ and $-14/3$ (Fig. 1c). High-field measurements as a function of carrier density reveal more details about the FQH states in the $N = 2$ LL. When sweeping the back-gate voltage at $B = 30$ T, we observe the states with denominator 3, as well as the more weakly formed 22/5, 23/5, 27/5 and 28/5 states (Fig. 2b). The $-13/3$ and $-14/3$ states, seen in device 1 at 30 T on the hole side (Fig. 2a), have analogues at $+13/3$ and $+14/3$ on the electron side in device 2 at fields as low as 7 T (Fig. 2d). A Landau fan diagram from device 3 shows the persistence of the $-13/3$ and $-14/3$ states from 7 to 14 T (Fig. 2c). We confirmed the assignment of these features to FQH states with both $R_{xx}$ and $R_{xy}$ data. In the low-field $R_{xx}$ data as a function of filling factor (normalized carrier density), these states appear as vertical lines, supporting their assignment as quantum Hall states (Fig. 2d). Quantization of the fractional $R_{xy}$ plateaux with denominator 3, when they are clearly discerned, is within 1% of $(1/v)(h/e^2)$ (Fig. 2a).

To gain insight into the nature of the ground states of the FQH states for $4 < |v| < 6$, we performed tilted magnetic-field measurements, which allow discrimination between the effects of Coulomb interaction (tuned by perpendicular field) and those of the Zeeman splitting (tuned by total field). We compared $R_{xx}$ and $R_{xy}$ measured at a perpendicular (total) field of 25 T with a measurement carried out at the same perpendicular field with an in-plane component of $\sim 37$ T (Supplementary Note 3 and Supplementary Fig. 4). The $-13/3, -14/3, -16/3$ and $-17/3$ states show little $R_{xx}$ minima variation, suggesting that these states are spin-polarized at these fields.

The sequence of filling factors of FQH states that we see in the $N = 2$ LL appears consistent with the CF model's accounting. In each of the first three fully symmetry-broken LLs within the $N = 2$ orbital LL, we see 1/3 and 2/3 states, and in the first two, we see 2/5 and 3/5 more weakly than the states with denominator 3, as expected in the CF framework and as seen in the $N = 0$ LL in GaAs. We do not see 1/5 or 4/5 in any LL. The 19/3 and 20/3 states that we observe in device 2 (Fig. 2b) are the highest observed, to the best of our knowledge, within-LL particle–hole symmetric pairs reported in any quantum Hall system; we do not observe any FQH states for $v > 7$. These essentially particle–hole symmetric results are unexpected in light of recent experimental findings of particle–hole asymmetry in the lowest LL in bilayer graphene.

**Measurements of the FQH gaps.** The magnitude of the energy gaps of FQH states is a measure of their stability. From the strongly temperature-dependent $R_{xx}$ (Fig. 3a showing data for

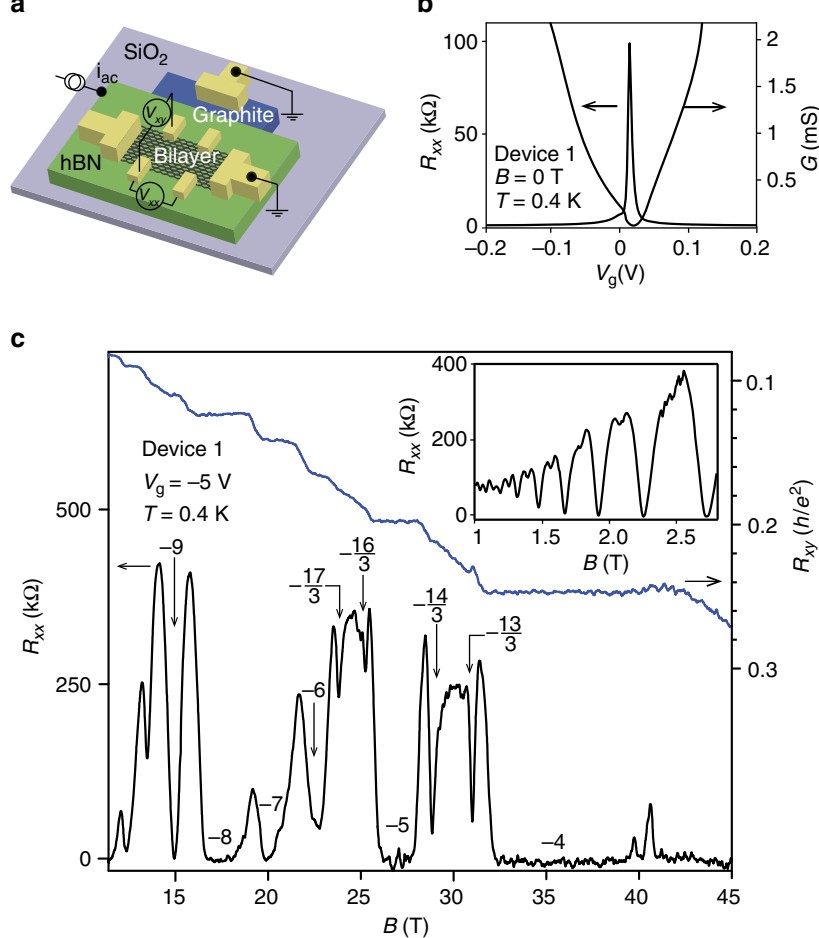

**Figure 1 | Device schematic and transport characteristics. (a)** Schematic of our bilayer graphene device design. **(b)** Zero-field resistance $R_{xx}$ and its inverse, conductance $G$, as a function of graphite back-gate voltage $V_g$ for device 1 (optical image in Supplementary Fig. 1). **(c)** Magnetoresistance $R_{xx}$ and Hall resistance $R_{xy}$ as a function of magnetic field $B$ at $V_g = -5$ V. Corresponding Landau level filling factors are labelled. Inset: low-field magnetoresistance with Shubnikov-de Haas oscillations showing the onset of degeneracy breaking among the integer states.

$4 < |v| < 5$ from device 3 on the hole side and Supplementary Fig. 6 for $4 < |v| < 6$ for device 2 on the electron side), we observe that the fractional states are largely suppressed above $\sim 3$ K (Supplementary Note 4). We extracted the activated energy gaps of four fractional states with denominator 3 ($-13/3$, $-14/3$) for $-5 < v < -4$ (device 3) and $4 < v < 6$ (device 2) (Supplementary Fig. 7) at several magnetic fields. The temperature-dependence of the $R_{xx}$ minima for the $-13/3$ and $-14/3$ states at 14 T fits the usual Arrhenius law $R_{xx} \propto e^{-\Delta/(2T)}$, with $\Delta$ the FQH energy gap divided by the Boltzmann constant and $T$ the temperature (Fig. 3b). Based on the fits for the data at $B = 14$ T, we calculate $\Delta_{-13/3} = (2.6 \pm 0.1)$ K (in units of Coulomb energy, $\sim 0.01$ $e^2/\varepsilon l_B$) and $\Delta_{-14/3} = (7.9 \pm 0.4)$ K ($\sim 0.03$ $e^2/\varepsilon l_B$), and for the states at 16/3 and 17/3, $\Delta_{16/3} = (7.5 \pm 0.2)$ K ($\sim 0.032 e^2/\varepsilon l_B$) and $\Delta_{17/3} = (7.0 \pm 0.6)$ K ($\sim 0.025 e^2/\varepsilon l_B$) where $l_B = (\hbar/eB)^{1/2}$ is the magnetic length.

Measured FQH gaps are normally significantly reduced by disorder broadening[28] and LL mixing[29]. In monolayer graphene[15] and the $N = 1$ LL in GaAs/AlGaAs systems[30] measured FQH gaps are at least one order of magnitude smaller than values predicted in the absence of disorder. For states that follow the expected CF sequence, the activation gap of particle–hole conjugate states is expected to be the same, as seen in the single activation energy [3]$\Delta$ measured for $v = 1/3$ and $2/3$ and [5]$\Delta$ for 2/5 and 3/5 in GaAs[31]. Therefore, assuming disorder equally affects the

particle–hole conjugate states $v^\star$ and its conjugate $1 - v^\star$, where $v^\star = 1/3$ or 2/5, the relative magnitude of the true activation gaps should track that of the experimentally measured gap sizes. Consistent with this expectation, $\Delta_{16/3}$ is close in value to $\Delta_{17/3}$ across the magnetic field range used in our study of activated gaps. In contrast, $\Delta_{-14/3}$ is larger than $\Delta_{-13/3}$ across the field range and is almost three times larger at 14 T. This difference could be due to LL mixing, which may be more significant for $-13/3$ than $-14/3$, decreasing $\Delta_{-13/3}$. Another possible cause is proximity to a transition in quantum numbers of the partially filled LL: in the Landau fan of Fig. 2c, the $v = -5$ gap is seen to weaken and then re-emerge as magnetic field is tuned through 9–10 T ($V_g \sim -1.8$ V).

## Discussion

In the $N = 2$ LL in bilayer graphene, we observe a sequence of FQH states that appears to be consistent with the accounting of the conventional CF model, including particle–hole symmetry. These results are significant and unexpected in light of recent experimental findings of particle–hole asymmetry in the lowest LL in bilayer graphene, prompting us to compare bilayer graphene FQH states in the $N = 0/N = 1$ LLs with those in the $N = 2$ LL. Given that the FQH effect does not survive in high LLs in GaAs-based systems, we also assess the applicability of the

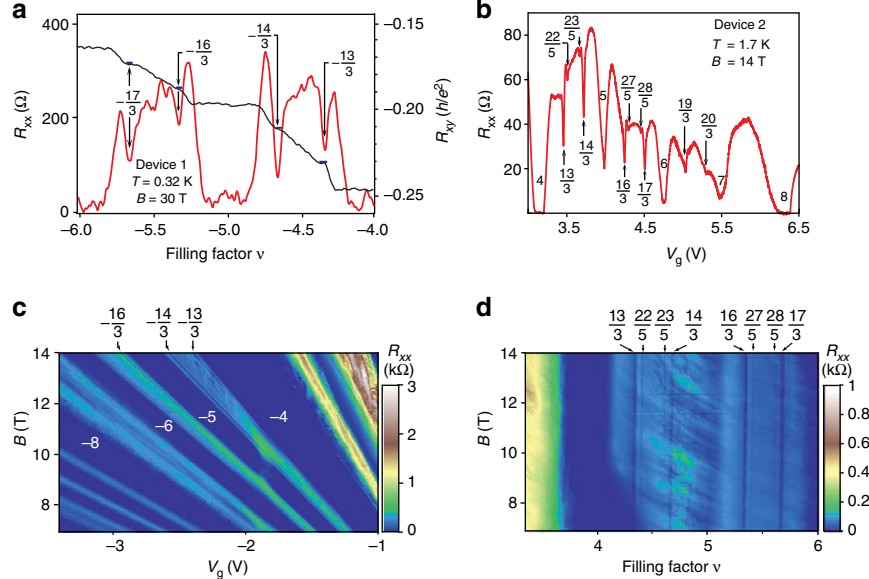

**Figure 2 | Particle–hole symmetric fractional quantum Hall effect in the $N=2$ Landau level.** (**a**) Longitudinal magnetoresistance $R_{xx}$ and Hall resistance $R_{xy}$ of device 1 at 30 T showing pronounced fractional states. (**b**) Fractional states seen in $R_{xx}$ on the electron side of device 2 at 14 T. (**c**) Landau fan diagram of $R_{xx}$ as a function of magnetic field and carrier density on the hole side for device 3. (**d**) $R_{xx}$ as a function of filling factor (carrier density rescaled by magnetic field) for device 2 on the electron side. Vertical features mark FQH states.

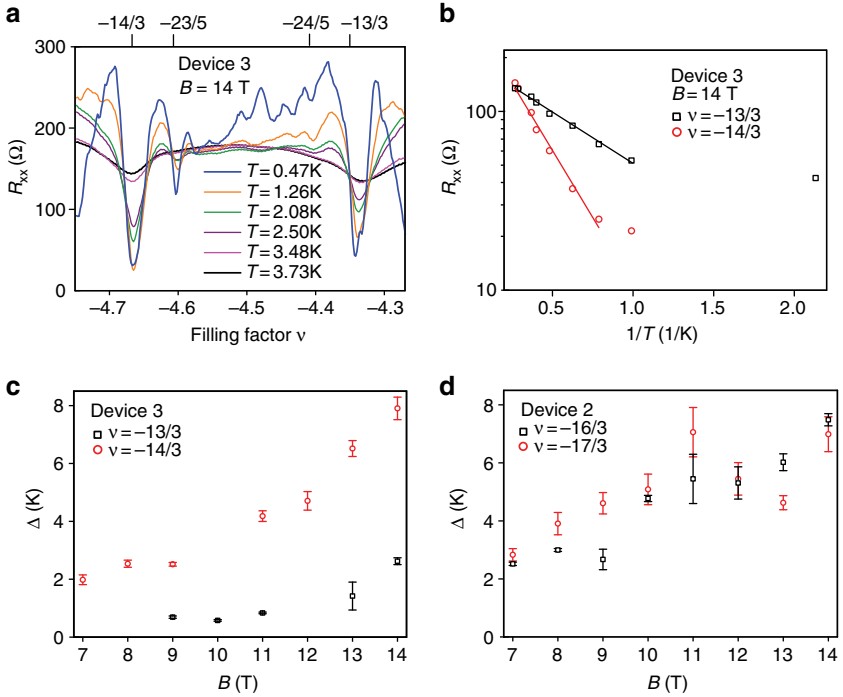

**Figure 3 | Fractional quantum Hall gaps in the $N=2$ Landau level.** (**a**) Temperature dependence of the magnetoresistance $R_{xx}$ for device 3 for $-5 < \nu < -4$ at 14 T, showing that the $R_{xx}$ minima for the states with denominator 3 deepen with decreasing temperature down to $\sim 1\,\mathrm{K}$. (**b**) $R_{xx}$ at $\nu = -13/3$ and $-14/3$ (device 3) plotted on a semilogarithmic scale as a function of inverse temperature. The linear fits yield activation gaps, greater for $\nu = -14/3$. The lowest temperature data points depart from activated behaviour, as is typically seen in QH systems at the onset of variable-range hopping and stronger localization. (**c**) Measured gaps as a function of magnetic field for $\nu = -13/3$ and $-14/3$ (device 3) and (**d**) for $\nu = 16/3$ and 17/3 (electron side in device 2). The error bars are due to the statistical error in fitting the data to the Arrhenius law $R_{xx} \propto e^{-\Delta/(2T)}$.

CF model to the states we have observed in the $N=2$ LL of bilayer graphene.

In the same samples in which we measured states in the $N=2$ LL, we also observed particle–hole asymmetric FQH states

in the $N=0/N=1$ LLs that do not form a complete CF sequence. $R_{xx}$ and $R_{xy}$ (Supplementary Fig. 5a and Supplementary Fig. 5b) of device 4 were measured at 30 T as a function of back-gate voltage in the accessible density range

on the hole side, showing the states $-7/3$, $-8/3$ and $-11/3$ in the $N=0/N=1$ LL. The states $2/3$, $4/3$, $5/3$ and $8/3$ are seen in device 2 on the electron side (Supplementary Fig. 5c), confirming the general observation from all measured samples that the states in the lowest LL do not form a complete CF sequence, unlike those in the $N=2$ LL (Supplementary Note 5). Other than a possible intra-LL electron–hole conjugate state—a barely resolved $5/3$ state symmetric to the $4/3$ state—the rest of the states we observe display particle–hole asymmetry. We observe $2/3$, $4/3$ and $8/3$ in the $N=0/N=1$ LLs on both the electron and hole sides while the $7/3$ and $-5/3$ states are absent, suggesting that valley degeneracy has not been broken. We also measured the energy gaps for several FQH states in the $N=1/N=0$ LLs, such as $5/3$ ($\Delta \sim 2.5$ K), $4/3$ ($\Delta \sim 4$ K) and $8/3$ ($\Delta \sim 1.2$ K) (Supplementary Fig. 8), showing that FQH gaps in the $N=0/N=1$ LLs are smaller than those in the $N=2$ LL (Supplementary Note 6).

Although our results for the FQH states in the $N=2$ LL in bilayer graphene are consistent with the CF sequence and obey particle–hole symmetry, our finding should also be considered in light of an alternative model[7,32,33] motivated by the breakdown in a high LL of the approximations on which the CF model is based. As the CF model might be destabilized in this circumstance, an alternative framework based on braid subgroup hierarchy has been advanced[7,34]. In its simplest, single-loop form, it predicts FQH states at all the fifths (states with denominator 5) in the $N=2$ LL (e.g. $21/5$, $22/5$, $23/5$ and $24/5$ for $4<|\nu|<5$) and no thirds (states with denominator 3) in this LL, whereas we observe just those fifths expected from the CF model ($22/5$ and $23/5$), as well as all the thirds expected from the CF model. It is possible that the fifths with numerators 1 and 4 that we do not observe ($21/5$ and $24/5$) but that are expected in this framework are subsumed into integer plateaus, while the observed thirds could be explained at higher order. However, given that the observed thirds are stronger than the fifths in the $N=2$ LL, our results more simply lend themselves to a CF interpretation, which would naturally yield both the states we observe and their relative strengths.

Theory specific to graphene also predicts robust fractional states in the $N=2$ LL of bilayer graphene[26], consistent with our experimental data, setting this system apart from conventional semiconductor systems in which states in the $N=2$ LL are charge-ordered. Fractional states with denominators 3 and 5 in bilayer graphene have been predicted to be as strong in the $N=2$ LL as in lower LLs owing to a shorter-range pseudopotential than in the $N=2$ LL of other systems[35]. A larger set of stable FQH states is expected to be accessed in bilayer graphene compared with their monolayer counterparts by electrically tuning the layer asymmetry in bilayer graphene[36] though our sample design did not allow this.

The sharp contrast between our observations and prior studies of bilayer graphene may be related to differences between the heterostructures studied: notably, we designed our heterostructures as open-face, bottom-gated bilayers in order to enhance Coulomb interactions[15]. This points to the opportunity to rationally optimize van der Waals heterostructures to host desired FQH states. Adding a suspended top gate to our style of bilayer device[37] (Supplementary Fig. 9) should enable the application of a large electric field normal to the graphene plane in order to probe FQH states in both low[38] and high LLs in bilayer graphene, without weakening the Coulomb interactions that drive the FQH effect (Supplementary Note 7).

## Methods

**Device fabrication.** We fabricated open-face bilayer graphene samples sitting on atomically smooth h-BN layers. Briefly, a thin ($\sim 5-10$ nm) graphite sheet

exfoliated on a $SiO_2/n^{++}$ Si substrate was chosen to serve as a local bottom gate for each bilayer graphene sample. h-BN flakes tens of nanometre thick were separately exfoliated on thin ($\sim 60$ nm) polyvinyl alcohol films spin-coated on bare Si. A suitable h-BN flake was chosen using optical and atomic force microscopic imaging and was subsequently transferred onto a part of the bottom-gate graphite using polymethyl methacrylate transfer and then annealed in 10% $O_2$ in Ar at 500 °C (ref. 39). Bilayer graphene was then transferred on the h-BN sheets and Hall bars were fabricated. We made no attempt to rotationally align our bilayers with the underlying h-BN flakes and we saw no signs of secondary (superlattice) Dirac peaks and fractal quantum Hall states in our devices[40–42]. For $B>1.5$ T, the magnetic length $l_B=26/\sqrt{B}$ (in nm) is always shorter than the h-BN thickness (27 or 47 nm for all devices studied) so that the graphite back-gate suppresses potential fluctuations without substantially screening the short-range Coulomb interactions responsible for the FQH states. Fabrication of such open-face samples was successfully accomplished with both polymethyl methacrylate (wet) transfer[15] and polypropylene carbonate (PPC) on top of polydimethylsiloxane (PDMS) (dry) transfer[43].

**Measurements.** The experiments were performed in a cryogen-free dilution refrigerator and in a ${}^3$He cryostat using standard ac lock-in techniques. Measurements at fields $>14$ T were performed at the National High Magnetic Field Laboratory in Tallahassee, FL, USA.

**Data analysis.** Gap values and error bars are obtained by plotting $\ln R_{xx}$ versus $1/T$ as shown in Fig. 3c and fitting to a line. The reported gap $\Delta$ is half the slope of the linear fit (in units of Kelvin). The error bars are the standard error associated with the linear least squares fit.

**Data availability.** The data that support the findings of this study are available from the corresponding authors upon request.

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

## Acknowledgements

We thank Eli Fox for experimental help. We thank Allan MacDonald, Jainendra Jain and Tapash Chakraborty for theoretical discussions about FQH in higher LLs and Michael Zaletel, Zlatko Papic, Michael Peterson and Kiryl Pakrouski for such discussions and also sharing unpublished calculations. Jurgen Smet shared enlightening thoughts on FQH physics in high LLs in various material systems. Luis Balicas graciously commented on an earlier version of the manuscript. Some of the measurements were performed at National High Magnetic Field Laboratory, which is supported by the US National Science Foundation cooperative agreement no. DMR-1157490. Experiments were funded in part by the Gordon and Betty Moore Foundation through Grant GBMF3429 to D.G.-G. G.D. was supported partly by a clean-energy seed grant from the Precourt Institute at Stanford University and by a Stanford Graduate Fellowship. P.G. acknowledges a Stanford Graduate Fellowship. M.L. acknowledges support from Samsung and Stanford University. A.J.B. was supported by a Benchmark Stanford Graduate Fellowship. C.-T.L. was funded by the MOST, Taiwan (grant numbers MOST 103-2918-I-002-028 and MOST 103-2622-E-002 -031).

## Author contributions

G.D., F.A. and D.G.-G. conceived the experiment; G.D., P.G. and F.A. designed and fabricated the samples; T.T. and K.W. grew BN crystals used for the sample fabrication; G.D., C.-T.L., F.A., A.J.B., M.L., K.T., L.B. and D.G.-G. conducted the measurements; W.C. and J.J. assisted extensively with measurements at the NHMFL; the manuscript was written by G.D., C.-T.L. and D.G.-G. with input from all authors.

## Additional information

**Competing financial interests:** The authors declare no competing financial interests.

