## [Peer Review File · Nature Communications]

Reviewers' comments:

Reviewer #1 (Remarks to the Author):

In this manuscript, the authors report the observation of fractional quantum Hall states in the $N=2$ Landau level of bilayer graphene. The authors find that the sequence of FQHE states is particle-hole symmetric, which should be contrasted with previous works that found strong particle-hole asymmetry of FQHE in $N=0,1$ Landau levels of bilayer graphene.

This is an interesting experiment, although the results are perhaps not unexpected (previous theory works predicted exactly this behavior; as a side note, the authors can do a better job citing previous theory works).

The data is of high quality, and overall I believe the paper is up to the high standard of Nature Communications. One interesting theoretical prediction concerns the tunability of FQHE states in $N=2$ LL by perpendicular electric field. Did authors attempt to study how FQH states change when strong bias is applied? I believe that the system under consideration exhibits most interesting physical effects in this regime (e.g., phase transitions between different phases can be realized, and possibly even non-Abelian states can be induced).

It is likely that the states observed in this experiment are of the same nature as states in GaAs, but this system has a lot of potential due to its tunability. This experiment opens the door to studying FQH effect in a highly tunable setting. Therefore, I recommend the publication, but it would be helpful if the authors include a short overview of theoretical works and discuss the potential for future experiments (and in particular, explain why this system can eventually be richer than GaAs).

Reviewer #2 (Remarks to the Author):

In this article the authors present clear experimental results showing the presence of fractional quantum Hall states in $N=2$ Landau level (LL) in bilayer graphene. In contrast to recent experimental results of asymmetric fractional quantum Hall states in $N=0$ and 1 LLs, their results for $N=2$ LL are particle-hole symmetric and the energy gaps are almost twice as large as those in the $N=0$ and 1 LLs. Tilted magnetic field measurements show these fractional quantum Hall states in $N=2$ LL are spin polarized.

They fabricated open-face bilayer graphene sample on hexagonal boron nitride to improve homogeneity of electron density with the reduction of dielectric constant that enhances energy scale of the fractional quantum Hall states.

I think this deliberate sample fabrication is a key to observe symmetric fractional quantum Hall states in $N=2$ LL for the first time. The presence of particle-hole symmetric fractional quantum Hall states in $N=2$ LL with relatively large excitation gap is consistent with theoretical predictions of ideal system. I think this consistency is an important step to understand the physics of strongly correlated electrons in nearly ideal system of atomic layer. The fully spin polarized ground state in $N=2$ LL is also consistent with the theoretical prediction and this result will stimulate further study on more complicated ground state of $N=0$ and 1 LLs where 8-fold degeneracy of spin, valley, and orbital degrees of freedom is present.

The quality of data is enough to assure the reliability of conclusions and explanations of the results are sound.

In my opinion, it is important to clarify the role of valley degrees of freedom for the

understanding of unique features of graphene and related materials. Theoretical reason for the relatively large excitation gap in $N=2$ LL is the absence of valley skyrmions, and the particle-hole asymmetry in $N=0$ and 1 LLs may be related to sublattice asymmetry of samples that makes asymmetry in valley degrees of freedom.

In this sense it is informative to clearly show the particle-hole asymmetry in $N=0$ and 1 LLs in the same device (#2 or #4), because only the electron or hole side is presented in Fig.S5.

In conclusion, I recommend this article for publication.

Reviewer #3 (Remarks to the Author):

The paper concerns the extremely interesting question of understanding of FQHE hierarchy structure in bilayer graphene. It is previously shown experimentally that in the LLL in bilayer graphene the unconventional states with even denominators have been observed [Nano Lett. 14, 2135-2139 (2014) doi: 10.1021/nl5003922] with the most pronounced feature at $-1/2$. This observation is clearly beyond the explanation ability of the conventional CF model. Therefore the question on how looks the FQHE hierarchy in $N=2$ LL of bilayer graphene is of primary importance. The paper answers to this question presenting very precise measurement of fractional Hall features in $N=2$ bands. The strongly stable states at $13/3$, $14/3$ and $16/3$, $17/3$ have been observed similarly as in higher band $19/3$, $20/3$. Even though the experimental effort and measurement quality is excellent and certainly deserves publication, the formulated conclusions are in our opinion misleading or at least not complete and must be improved.

1) The presence of observed in experiment twin doublets of states (as listed above) do not confirm the CF model; CF fermion hierarchy utilized to explanation of FQHE filling structure in the LLL of conventional semiconductor 2DEG has the main form of $n/[(p-1)n+1]$, n -integer, p -odd integer and is not observed in higher LLs; similarly in bilayer graphene the series $n/[(p-1)n+1]$ is also not observed for $N=2$; two twin doublets of states with the denominator 3 are not a manifestation of the CFs.

2) The presence of twin doublets with denominator 3 in the $N=2$ LL has been explained quite distinctly than the CF model with simultaneous elucidation why the full hierarchy from the LLL is not repeated in higher LLs.

Therefore, the assertion that the observed features in $N=2$ LL in bilayer graphene confirm CF model is not justified and must be reformulated in view of actual experimental observations which in fact rather contradict than confirm the CF model in higher LLs. Some hints according to the failure of CF model in higher LLs are recently published [JETP Letters 102, 19-25 (2015) doi: 10.1134/S0021364015130044; Proc. R. Soc. A 472, 20150330 (2016) doi: 10.1098/rspa.2015.0330; Scientific Reports 5, 14287-1-16 (2015) doi: 10.1038/srep14287].

Despite some controversy in interpretation, the paper is excellent and demonstrates the important progress in measurement technique of Hall states in graphene. This is the strong argument supporting its publication.

Response to Referee 1

1. Relation to prior experiments and theory.

In this manuscript, the authors report the observation of fractional quantum Hall states in the $N=2$ Landau level of bilayer graphene. The authors find that the sequence of FQHE states is particle-hole symmetric, which should be contrasted with previous works that found strong particle-hole asymmetry of FQHE in $N=0,1$ Landau levels of bilayer graphene.

This is an interesting experiment, although the results are perhaps not unexpected (previous theory works predicted exactly this behavior; as a side note, the authors can do a better job citing previous theory works).

We thank Referee 1 for useful comments and enthusiasm. In accordance with the Referee's suggestions, we have now cited more previous theory papers (including papers addressing the fractional quantum Hall effect in the lowest Landau level) and added associated text:

Added to revised paper (line 8):

“In addition to the CF model, a hierarchical scheme of parent and daughter states^{5,6} and a model based on cyclotron braid subgroups⁷ offer competing explanations for the pronounced FQH states at $\nu = p/(2mp \pm 1)$ in the lowest LL.”

In revised paper (line 26):

“Despite the consistency of the states we observe with the accounting of the CF model, we discuss the possibility of their true microscopic description as non-CF states by alluding to states in high ($N \geq 2$) LLs of GaAs-based 2D electron systems. In the latter, aside from possible evidence for $\nu = 4+1/5$ and $\nu = 4+4/5$ FQH states²⁰, competing charge-ordered states such as Wigner crystal bubbles and nematic stripes are thought to be the many-body ground states^{21–23} in the $N=2$ LL (filling factors $4 < \nu < 6$). Why might charge-ordered states be expected to supplant FQH and specifically CF states in high LLs? In high ($N > 0$) LLs, the more extended electron wave-functions give rise to greater residual interactions between CFs²⁴ and may destabilize the FQH states^{21–23, 25}. Such wave-functions have nodes at particular momenta corresponding to spatial separation between orbitals on the order of the cyclotron radius and favoring charge ordering with that spacing²⁶. We also highlight an alternative explanation that supports the

validity of the CF model to our results - predictions of unique to graphene features of LL wave-functions that may stabilize CF states in the $N=2$ LL of bilayer graphene.”

2. Data in perpendicular electric field?

The data is of high quality, and overall I believe the paper is up to the high standard of Nature Communications. One interesting theoretical prediction concerns the tunability of FQHE states in $N=2$ LL by perpendicular electric field. Did authors attempt to study how FQH states change when strong bias is applied? I believe that the system under consideration exhibits most interesting physical effects in this regime (e.g., phase transitions between different phases can be realized, and possibly even non-Abelian states can be induced).

We thank the Referee for positive comments on data quality. Though we are aware of both theory and other experiments studying the rich quantum Hall landscape accessed in bilayer graphene in perpendicular electric field, our devices are by design all back-gated and do not have additional top gates. This is to keep the dielectric constant low (to enhance the effect of interactions) and to avoid some of the contact challenges in studies of encapsulated graphene in quantum Hall regime.

Referee 2 identified “*this deliberate sample fabrication [as] a key to observe symmetric fractional quantum Hall states in $N=2$ LL for the first time.*” With our type of structure, although we are able to apply a strong bias on the back-gate, we cannot obtain a large potential difference between the graphene layers, a requirement for observing transitions between different phases and non-Abelian FQH states. In future experiments we plan to add a suspended top gate in order to keep the effective dielectric constant low so that the Coulomb interactions are strong. Such a device will allow us to tune both total density in the two layers and potential difference between the layers, to probe phenomena such as phase transitions and non-Abelian states identified by the Referee as worthy targets. Since the Referee also mentioned future experiments and an issue regarding why bilayer graphene is richer than GaAs from the view point of studying the fractional quantum Hall effect, please see our reply in point 3.

3. Comparison to GaAs: theory and future experiments

It is likely that the states observed in this experiment are of the same nature as states in GaAs, but this system has a lot of potential due to its tunability. This

experiments opens the door to studying FQH effect in a highly tunable setting. Therefore, I recommend the publication, but it would be helpful if the authors include a short overview of theoretical works and discuss the potential for future experiments (and in particular, explain why this system can eventually be richer than GaAs).

We appreciate that the Referee recommends publication of our paper and makes the excellent suggestion to add a short overview of theoretical works and the potential for future experiments to illustrate how FQH in bilayer graphene can be richer than in GaAs.

In accordance with the Referee's suggestions, we have added discussion of theoretical works on the fractional quantum Hall effect in graphene-based systems. This is now followed by a description of the potential for future experiments and the tunability of both top- and back-gated bilayer graphene to access richer (and different) FQH in this system than in GaAs.

In revised paper (line 145):

“While our results for the FQH states in the $N=2$ LL in bilayer graphene conform to the expected CF sequence and particle-hole symmetry, our findings should also be considered in light of alternative models^{7,32,33} motivated in part by expected strong residual CF-CF interactions²⁴. Since the CF model might be destabilized in this circumstance, an alternative framework based on braid subgroup hierarchy has been advanced^{7,34} In its simplest, single-loop form it predicts FQH states at all the “fifths” (states with denominator 5) in the $N=2$ LL (e.g. $21/5$, $22/5$, $23/5$, and $24/5$ for $4 < |\nu| < 5$) and no “thirds” (states with denominator 3) in this LL, whereas we observe just those fifths expected from the CF model ($22/5$ and $23/5$), as well as all the thirds expected from the CF model. It is possible that the fifths with nominator 1 and 4 that we do not observe ($21/5$ and $24/5$) but that are expected in this framework are subsumed into integer plateaux, while the observed thirds could be explained at higher order. However, given that the thirds are stronger than the fifths in the $N=2$ LL, our results more simply lend themselves to a CF interpretation, which would naturally yield both the states we observe and their relative strengths.”

In revised concluding paragraph (line 173):

“Adding a suspended top gate to our style of bilayer device³⁸ (Supplementary Fig. 9) should enable the application of a large electric field normal to the graphene plane in order to probe FQH states in both low and high Landau levels in bilayer graphene, without weakening the Coulomb interactions that drive the FQH effect.”

Response to Referee 2

4. Impact and data quality

In this article the authors present clear experimental results showing the presence of fractional quantum Hall states in $N=2$ Landau level (LL) in bilayer graphene. In contrast to recent experimental results of asymmetric fractional quantum Hall states in $N=0$ and 1 LLs, their results for $N=2$ LL are particle-hole symmetric and the energy gaps are almost twice as large as those in the $N=0$ and 1 LLs. Tilted magnetic field measurements show these fractional quantum Hall states in $N=2$ LL are spin polarized.

They fabricated open-face bilayer graphene sample on hexagonal boron nitride to improve homogeneity of electron density with the reduction of dielectric constant that enhances energy scale of the fractional quantum Hall states. I think this deliberate sample fabrication is a key to observe symmetric fractional quantum Hall states in $N=2$ LL for the first time. The presence of particle-hole symmetric fractional quantum Hall states in $N=2$ LL with relatively large excitation gap is consistent with theoretical predictions of ideal system. I think this consistency is an important step to understand the physics of strongly correlated electrons in nearly ideal system of atomic layer. The fully spin polarized ground state in $N=2$ LL is also consistent with the theoretical prediction and this result will stimulate further study on more complicated ground state of $N=0$ and 1 LLs where 8-fold degeneracy of spin, valley, and orbital degrees of freedoms is present.

The quality of data is enough to assure the reliability of conclusions and explanations of the results are sound.

... In conclusion, I recommend this article for publication.

We thank Referee 2 for positive comments and recommendation to publish our manuscript in Nature Communications.

5. Valley degree of freedom.

In my opinion, it is important to clarify the role of valley degrees of freedoms for the understanding of unique features of graphene and related materials. Theoretical reason for the relatively large excitation gap in $N=2$ LL is the absence of valley skyrmions, and the particle-hole asymmetry in $N=0$ and 1 LLs may be related to sublattice asymmetry of samples that makes asymmetry in valley degrees of freedom. In this sense it is informative to clearly show the particle-hole asymmetry in $N=0$ and 1 LLs in the same device (#2 or #4), because only the electron or hole side is presented in Fig.S5.

The Referee raises an important issue regarding the asymmetry of valley degrees of freedom. We agree that it is informative to show the particle-hole asymmetry on both the electron and hole sides. We have therefore added a figure (Fig. S5c) and two sentences to describe the possible asymmetry of the valley degree of freedom in the $N=0/N=1$ Landau levels in bilayer graphene.

In revised paper (line 143):

“We observe $2/3$, $4/3$ and $8/3$ in the $N=0/N=1$ LLs on both the electron and hole sides while the $7/3$ and $-5/3$ states are absent, suggesting that valley degeneracy has not been broken.”

Response to Referee 3

6. Good data; theoretical links need improvement

The paper concerns the extremely interesting question of understanding of FQHE hierarchy structure in bilayer graphene. It is previously shown experimentally that in the LLL in bilayer graphene the unconventional states with even denominators have been observed [Nano Lett. 14, 2135-2139 (2014) doi: 10.1021/nl5003922] with the most pronounced feature at $-1/2$. This observation is clearly beyond the explanation ability of the conventional CF model. Therefore the question on how looks the FQHE hierarchy in $N=2$ LL of bilayer graphene is of primary importance. The paper answers

to this question presenting very precise measurement of fractional Hall features in $N=2$ bands. The strongly stable states at $13/3$, $14/3$ and $16/3$, $17/3$ have been observed similarly as in higher band $19/3$, $20/3$. Even though the experimental effort and measurement quality is excellent and certainly deserves publication, the formulated conclusions are in our opinion misleading or at least not complete and must be improved. ... Despite some controversy in interpretation, the paper is excellent and demonstrates the important progress in measurement technique of Hall states in graphene. This is the strong argument supporting its publication.

We thank the Referee for remarking that the experimental effort and measurement quality is excellent, and that our paper deserves publication. The Referee raises an important issue regarding whether the composite fermion picture is valid in the $N=2$ Landau level which we address in point (7) below.

7. FQH states may not be CF

1) The presence of observed in experiment twin doublets of states (as listed above) do not confirm the CF model; CF fermion hierarchy utilized to explanation of FQHE filling structure in the LLL of conventional semiconductor 2DEG has the main form of $n/[(p-1)n+1]$, n -integer, p -odd integer and is not observed in higher LLs; similarly in bilayer graphene the series $n/[(p-1)n+1]$ is also not observed for $N=2$; two twin doublets of states with the denominator 3 are not a manifestation of the CFs.

2) The presence of twin doublets with denominator 3 in the $N=2$ LL has been explained quite distinctly than the CF model with simultaneous elucidation why the full hierarchy from the LLL is not repeated in higher LLs.

Therefore, the assertion that the observed features in $N=2$ LL in bilayer graphene confirm CF model is not justified and must be reformulated in view of actual experimental observations which in fact rather contradict than confirm the CF model in higher LLs. Some hints according to the failure of CF model in higher LLs are recently published [JETP Letters 102, 19-25 (2015) doi: 10.1134/S0021364015130044; Proc. R. Soc. A 472, 20150330 (2016) doi: 10.1098/rspa.2015.0330; Scientific Reports 5, 14287-1-16 (2015) doi: 10.1038/srep14287].

The Referee raises an important issue regarding the CF hierarchy: observation of QH states at any particular fractions does not prove that these states are properly described

by the CF picture.

We would like to state that in the $N=2$ LL in bilayer graphene, we observe a sequence of FQH states, with local minima of R_{xx} visible at factors $\nu + \delta\nu$, where $\nu = 4$ and 5 and $\delta\nu = 1/3, 2/3, 2/5$ and $3/5$, as well as the $6+1/3$ and $6+2/3$ states. Such a FQH sequence is both particle-hole conjugate within each fully symmetry-broken LL, and follows the same sequence as composite fermions. Moreover, the observation of $2/5$ and $3/5$ states, weaker than the thirds but still clear, fits the expectation of the CF model, where these states are the next-most prominent in the hierarchy. At the same time, we do not observe the $1/5$ and $4/5$ states, which are not part of the CF description.

Nonetheless, we agree that the observation of states fitting the CF sequence does not prove that these are CF states. Alternative ground states have been proposed at some of these filling factors, motivated in part by expectations of strong residual CF-CF interactions (Töke *et al.*, Ref. 24 in the revised manuscript) and the resulting failure of the CF model in high Landau levels (JETP Letters 102, 19 (2015), Proc. R. Soc. A 472, 20150330 (2016) and Scientific Reports 5, 14287 (2016)). Moreover, it is evident that the CF model cannot be used to explain the $4+1/5$ and $4+4/5$ states, the only two FQH states for which evidence has been offered to date in the $N=2$ Landau level in GaAs (Ref. 20).

Inspired by the Referee's comments, we have revised our paper as listed below:

The title of our original paper was:

“Robust fractional quantum Hall effect and composite fermions in the $N=2$ Landau level in bilayer graphene”.

The new title reads:

“Robust fractional quantum Hall effect in the $N=2$ Landau level in bilayer graphene: composite fermions?”

In our original Abstract:

“In contrast with recent observations of particle-hole asymmetry in the $N=0/N=1$ LLs of bilayer graphene, the FQH states we observe in the $N=2$ LL are consistent with the CF model: within a LL, they form a complete sequence of particle-hole

symmetric states whose relative strength is dependent on their denominators.”

In revised Abstract:

“In contrast with recent observations of particle-hole asymmetry in the $N=0/N=1$ LLs of bilayer graphene, the FQH states we observe in the $N=2$ LL follow the CF sequence. Although predictions exist for the enhanced stability of high-LL FQH states in bilayer graphene, the validity of the generic CF model in high LLs is challenged by the expected strong residual CF-CF interactions. The competing arguments of the CF model and alternative theories are discussed.”

In the original paper:

“Here, we report observations of particle-hole symmetric FQH states, consistent with a composite fermion accounting, in the $N=2$ LL in bilayer graphene.”

In revised paper (line 25):

“Here, we report observations of particle-hole symmetric FQH states in the $N=2$ LL in bilayer graphene. Despite the consistency of the states we observe with the accounting of the CF model, we discuss the possibility of their true microscopic description as non-CF states by alluding to states in high ($N \geq 2$) LLs of GaAs-based 2D electron systems. In the latter, aside from possible evidence for $\nu = 4+1/5$ and $\nu=4+4/5$ FQH states²⁰, competing charge-ordered states such as Wigner crystal bubbles and nematic stripes are thought to be the many-body ground states²¹⁻²³ in the $N=2$ LL (filling factors $4 < \nu < 6$). ... We also highlight an alternative explanation that supports the validity of the CF model to our results - predictions of unique to graphene features of LL wave-functions that may stabilize CF states in the $N=2$ LL of bilayer graphene.”

In the original paper:

“Quantisation of the R_{xy} plateau, when they are clearly discerned, is within 1% of $(1/\nu)(h/e^2)$. The FQH states we observe in the $N=2$ LL in bilayer graphene follow the composite fermion sequence with $\nu = p/(2p\pm 1)$ yielding particle-hole conjugate states.”

In revised paper (line 79):

“The sequence of filling factors of FQH states that we see in the $N=2$ LL appears consistent with the CF model’s accounting. In each of the first three parent integer filling factors within the fully symmetry-broken $N=2$ LL, we see $1/3$ and $2/3$ states, and in the first two, we see $2/5$ and $3/5$ more weakly than the states with denominator 3, as expected in the CF framework and as seen in the $N=0$ LL in GaAs. We do not see $1/5$ or $4/5$ in any LL. The $19/3$ and $20/3$ states that we observe in device 2 (Fig. 2b) are the highest observed, to the best of our knowledge, within-LL particle-hole symmetric pairs reported in any quantum Hall system; we do not observe any FQH states for $\nu > 7$. These essentially particle-hole symmetric results are unexpected in light of recent experimental findings of particle-hole asymmetry in the lowest LL in bilayer graphene.”

In accordance with the Referee’s comments, we have also cited pertinent theories on the fractional quantum Hall effect for the lowest Landau level as well as those in graphene-based systems. In particular, we mention the braid subgroup theory of Jacak and Jacak.

Reviewers' comments:

Reviewer #2 (Remarks to the Author):

In the revised manuscript the authors include a brief explanation of the FQH state in 2D systems mainly based on the CF model. The explanation using the CF theory is in some sense clear, but we should be careful to the limitation of the CF theory. This theory is based on a mean-field approximation. In this sense, the sentence in the abstract "the FQH states we observe in the $N=2$ LL follows the CF sequence, even though the CF model might not have been expected to be valid given strong residual CF-CF interactions." is inappropriate because the main purpose of the present article is to show the experimental results, and the validity of the CF theory should be studied within a given theoretical model. Indeed, the residual CF-CF interactions should be weak when the ground state is incompressible liquid with a large excitation gap. The question whether the electrons in bilayer graphene are really described by multi-component wave function of Dirac particles is more important, and it is constructive to mention the consistency with unbiased numerical works that do not use mean-field type approximations.

I would like to stress again that the experimental results of the present paper are important and worth publishing.

Reviewer #3 (Remarks to the Author):

We have noticed that the argumentation of our previous comment is only partly taken into account. In the corrected version of the submission, the Authors write: lines 152-155, "While our results for the FQH states in the $N=2$ LL in bilayer graphene conform to the expected CF sequence and particle-hole symmetry, our findings should also be considered in light of alternative models^{7,33,34} motivated in part by expected strong residual CF-CF interactions²⁴."

It is highly exaggerated to claim that 'the results conform to the expected CF sequence'. It is not justified such a claim only on the basis of the observation of $1/3$ family states for $N=2$ level in bilayer graphene. Why other CF hierarchy states are absent? The answer is that the observed states are not CFs, similarly as e.g., $1/2$ state in the LLL in bilayer Hall system is also not CF. The realistic explanation of FQHE hierarchy (including bilayer systems) avoiding the CF notion is given by braid cyclotron subgroups. CFs are only effective phenomenological model of braid group cyclotron commensurability at some exceptional conditions, which do not meet, however, in bilayer graphene. Some artificial concept as 'strong residual CF-CF interaction' only worsens situation by proposing additional phenomenological factor, 'residual strong interaction', which is not helpful in bilayer system, on the other hand. Moreover, the approach of interacting CF-CF to out-of-main-hierarchy states (e.g., $5/13$ in the LLL) is still vague and probably incorrect. This idea has nothing in common with cyclotron braid group commensurability approach. Thus the phrase:

"... motivated in part by expected strong residual CF-CF interactions²⁴."

is misleading and should be reformulated in agreement with true ability of the (confusing in fact) concept of CF-CF interaction. The hypothetical residual interaction of CF-CF which suddenly may occur at some fractions is odd for CF-quasiparticles non-interacting at other filling fractions.

**Reply to Second Round of Referee Reports, NCOMMS-16-05248B, Diankov *et al.*
August 12, 2016**

0. Referee 1

Though Referee 1 did not respond to our initial resubmission, that Referee had already strongly indicated that our work should be published in Nature Communications, upon minor local edits which we already implemented in our initial resubmission.

Response to Referee 2

1. Impact of experimental data

I would like to stress again that the experimental results of the present paper are important and worth publishing.

We thank Referee 2 for his/her enthusiasm for our results.

2. Emphasizing data over comparison between experimental data and theory.

In the revised manuscript the authors include a brief explanation of the FQH state in 2D systems mainly based on the CF model. The explanation using the CF theory is in some sense clear, but we should be careful to the limitation of the CF theory. This theory is based on a mean-field approximation. In this sense, the sentence in the abstract "the FQH states we observe in the $N=2$ LL follows the CF sequence, even though the CF model might not have been expected to be valid given strong residual CF-CF interactions." is inappropriate because the main purpose of the present article is to show the experimental results, and the validity of the CF theory should be studied within a given theoretical model. Indeed, the residual CF-CF interactions should be weak when the ground state is incompressible liquid with a large excitation gap. The question whether the electrons in bilayer graphene are really described by multi-component wave function of Dirac particles is more important, and it is constructive to mention the consistency with unbiased numerical works that do not use mean-field type approximations.

We thank Referee 2 for useful comments. We agree with the Referee that we should acknowledge the limitations of the composite fermion theory and we thank the Referee for sharing qualitative reasoning regarding residual CF-CF interactions. We also agree with the Referee that unbiased numerical works that do not use mean-field type approximations should be mentioned in our paper. To date, most theoretical studies on the FQH states on bilayer graphene are focused on the case when a large bias is applied across the bilayer. To the best of our knowledge, the pioneering work by Shibata and Nomura is the only published numerical paper on the $N=2$ Landau level. They made a prediction of the energy gaps in bilayer graphene which we already cited. In the revised manuscript we now emphasize their prediction of fractional quantum Hall states in the $N=2$ Landau level, and their description of possible physical origin of those states. We have also cited one other numerical work on fractions in bilayer graphene which does not use the mean-field approximation [Z. Papić and D. A. Abanin, Phys. Rev. Lett. 112, 046602 (2014)].

Inspired by the Referee's comments, we have revised our paper as listed below:

In original abstract:

“In contrast with recent observations of particle-hole asymmetry in the $N=0/N=1$ LLs of bilayer graphene, the FQH states we observe in the $N=2$ LL follow the CF sequence, even though the CF model might not have been expected to be valid given strong residual CF-CF interactions.”

In revised Abstract:

“In contrast with recent observations of particle-hole asymmetry in the $N=0/N=1$ LLs of bilayer graphene, the FQH states we observe in the $N=2$ LL obey particle-hole symmetry within the fully symmetry-broken LL.”

In original paper (line 36-38)

“In bilayer graphene, the wave functions in the $N=2$ LL have no complete nodes, and hence might be expected to support FQH states over charge-ordered states.”

In revised manuscript (line 36-41)

“In bilayer graphene, the wave functions in the $N=2$ LL have no complete nodes, and hence might be expected to support FQH states over charge-ordered states.²⁷ A numerical study which does not rely on the mean-field approximation or otherwise assume the CF picture predicts pronounced single-component FQH states at $1/3$, $2/3$, and $2/5$ (Ref. 27).”

In original paper (line 168-170)

“Theory specific to graphene also supports the observation of robust fractional states in the $N=2$ LL of bilayer graphene, setting this system apart from conventional semiconductor systems.”

In revised manuscript (line 171-174)

“Theory specific to graphene also predicts robust fractional states in the $N=2$ LL of bilayer graphene²⁷, consistent with our experimental data, setting this system apart from conventional semiconductor systems in which states in the $N=2$ LL are charge-ordered.”

Response to Referee 3

3. Residual CF-CF interactions

We have noticed that the argumentation of our previous comment is only partly taken into account. In the corrected version of the submission, the Authors write: lines 152-155, "While our results for the FQH states in the $N=2$ LL in bilayer graphene conform to the expected CF sequence and particle-hole symmetry, our findings should also be considered in light of alternative models^{7,33,34} motivated in part by expected strong residual CF-CF interactions²⁴."

.....Some artificial concept as 'strong residual CF-CF interaction' only worsens situation by proposing additional phenomenological factor, 'residual strong interaction', which is not helpful in bilayer system, on the other hand. Moreover, the

approach of interacting CF-CF to out-of-main-hierarchy states (e.g., 5/13 in the LLL) is still vague and probably incorrect. This idea has nothing in common with cyclotron braid group commensurability approach. Thus the phrase: "... motivated in part by expected strong residual CF-CF interactions²⁴." is misleading and should be reformulated in agreement with true ability of the (confusing in fact) concept of CF-CF interaction. The hypothetical residual interaction of CF-CF which suddenly may occur at some fractions is odd for CF-quasiparticles non-interacting at other filling fractions.].

We thank the Referee for useful comments on residual CF-CF interactions. As can be seen in point 2 above, we have removed the statement on residual CF-CF interactions in the abstract, so the abstract just describes the phenomenology of our observed FQH states rather than speculating on theoretical descriptions. In accordance with the Referee's comments, we have also revised our manuscript as follows:

In our original manuscript (line 152-155):

"While our results for the FQH states in the $N=2$ LL in bilayer graphene conform to the expected CF sequence and particle-hole symmetry, our findings should also be considered in light of alternative models^{7,33,34} motivated in part by expected strong residual CF-CF interactions"

In revised manuscript (line 155-159):

"While our results for the FQH states in the $N=2$ LL in bilayer graphene are consistent with the CF sequence and obey particle-hole symmetry, our finding should also be considered in light of an alternative model^{7, 33, 34} motivated by the breakdown of the approximations on which the CF model is based, in a high LL"

4. FQH states may not be CF

It is highly exaggerated to claim that 'the results conform to the expected CF sequence'. It is not justified such a claim only on the basis of the observation of 1/3 family states for $N=2$ level in bilayer graphene. Why other CF hierarchy states are absent? The answer is that the observed states are not CFs, similarly as e.g., 1/2 state in the LLL in bilayer Hall system is also not CF. The realistic explanation of FQHE hierarchy (including bilayer systems) avoiding the CF notion is given by braid cyclotron subgroups. CFs are only effective phenomenological model of braid group cyclotron commensurability at some exceptional conditions, which do not meet, however, in bilayer graphene.

We thank the Referee for these comments. The Referee is correct that the 1/2 state in the lowest Landau level in a bilayer quantum Hall system in GaAs is not CF. This 1/2 state arises from inter-layer interactions between a single 1/4-filled Landau level in each layer. However, this 1/2 state is not directly related to our experimental work, which may have led to the Referee's confusion. We would like to re-emphasize that in the $N=2$ LL in bilayer graphene, we observe a sequence of FQH states, with local minima of R_{xx} visible at factors $\nu + \delta\nu$, where $\nu = 4$ and 5 and $\delta\nu = 1/3, 2/3, 2/5$ and $3/5$, as well as the $6+1/3$ and $6+2/3$ states. Such a FQH sequence is particle-hole conjugate within each fully

symmetry-broken LL. Moreover, the observation of $2/5$ and $3/5$ states, weaker than the thirds but still clear in two of these three fully symmetry-broken LLs, fits the expectation of the CF model, where these states are the next-most prominent in the FQH hierarchy. We do not observe the $1/5$ and $4/5$ states, which are not part of the CF description.

Nonetheless, we agree with the Referee that the observation of states that fits the CF sequence does not prove that these are CF states. Alternative ground states have been proposed at some of these filling factors, motivated by the failure of the CF model in high Landau levels (JETP Letters 102, 19 (2015), Proc. R. Soc. A 472, 20150330 (2016) and Scientific Reports 5, 14287 (2016). Moreover, it is evident that the CF model cannot be used to explain the $4+1/5$ and $4+4/5$ states, the only two FQH states for which possible evidence has been offered to date in the $N=2$ Landau level in GaAs (Ref. 20) — though note that the authors of that work did not assign these as FQH states.

Inspired by the Referee's comments, we have revised our paper below.

In original manuscript (line 128-130)

“In the $N=2$ LL in bilayer graphene, we observe a sequence of FQH states that obeys the accounting of the conventional CF model, including particle-hole symmetry.”

In revised manuscript (line 131-133)

“In the $N=2$ LL in bilayer graphene, we observe a sequence of FQH states that appears to be consistent with the accounting of the conventional CF model, including particle-hole symmetry.”

In the revised manuscript, we have also addressed the applicability of the CF model raised by both Referee 2 and Referee 3. Please see point 2 above.

REVIEWERS' COMMENTS:

Reviewer #2 (Remarks to the Author):

In the newly revised version of the manuscript, proper notice is given.

The present paper shows the consistency between experiment and theory that confirms the correctness of the model Hamiltonian and the realization of interacting Dirac particles in a nearly ideal bilayer system.

I therefore recommend this paper for publication.

Reviewer #3 (Remarks to the Author):

Some additional note might be of order. The Authors suggested that CF picture might be stabilized due to absence of nodes in the wave function for bilayer graphene at $N=2$, as written in lines 30-41 of the manuscript,

"Why might

31 charge-ordered states be expected to supplant FQH and specifically CF states in high
32 LLs? In high ($N>0$) LLs, the more extended electron wave-functions give rise to
33 greater residual interactions between CFs²⁴ and may destabilize the FQH states21-
34 23,25. In GaAs such wavefunctions have nodes at particular momenta corresponding
35 to spatial separation between orbitals on the order of the magnetic length and
36 favoring charge ordering with that spacing²⁶. In bilayer graphene, the wave
37 functions in the $N=2$ LL have no complete nodes, and hence might be expected
38 to support FQH states over charge-ordered states.²⁷ A numerical study which
39 does not rely on the mean-field approximation or otherwise assume the CF
40 picture predicts pronounced single-component FQH states at $1/3$, $2/3$, and $2/5$
41 (Ref. 27)."

Even though the above observation on the absence of nodes in the wave function in bilayer graphene for $N=2$ is interesting, the suggested conclusion might be missing. The problem is that, despite the common opinion, CFs are not quasiparticles in the sense of dressing of bare electrons with the interaction. This is caused by the fact highlighted by Laughlin and called by him as 'quantization of electron separation in FQHE'. The related discontinuity of the mass operator generated by the interaction precludes a definition of quasiparticles in the system corresponding to FQHE (to define Landau type quasiparticles the continuity of the mass operator is unavoidably needed). Hence, the CFs are only heuristic construction which displays the multiparticle correlations in the planar charged system at magnetic field presence, and hypothetical CF residual interaction is not derived from the electron repulsion upon any formalism. The interaction is, however, crucial for the 2D topological-type multiparticle correlation creation. The FQHE correlation manifests itself at some specific filling ratios displaying topology constraints in 2D at magnetic field resulting in energy gain over other possible multiparticle states (Wigner crystal insulator, charge density wave, etc). The concept that hypothetical quasiparticles CFs appear or disappear, residually interact or not, due to small shift of density apparently evidences that such quasiparticles actually do not exist but are a convenient representation of topological correlation at specific filling ratios, though with some limits for the usability of the CF picture, especially in higher LLs.

Even though the statement of the authors in lines 30-41 might be vague to some extent, I do not insist for its modification because it is still interesting and worth mentioning.

The other modification of the paper is satisfactory. Thus I suggest to publish it in the present form.

0. Referee 1.

Though Referee 1 did not respond to our initial resubmission, that Referee had already strongly indicated that our work should be published in Nature Communications, upon minor local edits which we already implemented in our initial resubmission.

Response to Referee 2

1. Satisfactory revisions

In the newly revised version of the manuscript, proper notice is given. The present paper show the consistency between experiment and theory that confirms the correctness of the model Hamiltonian and the realization of interacting Dirac particles in a nearly ideal bilayer system. I therefore recommend this paper for publication.

We thank Referee 2 for his/her enthusiasm for our results.

Response to Referee 3

2. Suitable modification

The other modification of the paper is satisfactory. Thus I suggest to publish it in the present form.

We thank Referee 3 for his/her enthusiasm for our results expressed earlier, and for deciding that our modifications addressed his/her primary concerns.

3. Composite fermions and residual CF-CF interactions

Some additional note might be of order. The Authors suggested that CF picture might be stabilized due to absence of nodes in the wave function for bilayer graphene at $N=2$, as written in lines 30-41 of the manuscript, [“Why might charge-ordered states be expected to supplant FQH and specifically CF states in high LLs? In high ($N>0$) LLs, the more extended electron wave-functions give rise to greater residual interactions between CFs²⁴ and may destabilize the FQH states 21–23,25]. In GaAs such wavefunctions have nodes at particular momenta corresponding to spatial separation between orbitals on the order of the magnetic length and favoring charge ordering with that spacing²⁶. In bilayer graphene, the wave functions in the $N=2$ LL have no complete nodes, and hence might be expected to support FQH states over charge-ordered states.²⁷ A numerical study which does not rely on the mean-field approximation or otherwise assume the CF picture predicts pronounced single-component FQH states at $1/3$, $2/3$, and $2/5$ (Ref. 27).”]

Even though the above observation on the absence of nodes in the wave function in bilayer graphene for $N=2$ is interesting, the suggested conclusion might be missing. The problem is that, despite the common opinion, CFs are not quasiparticles in the sense of dressing of bare electrons with the interaction. This is caused by the fact highlighted by Laughlin and called by him as ‘quantization of electron separation in FQHE’. The related discontinuity of the mass operator generated by the interaction precludes a definition of quasiparticles in the system corresponding to FQHE (to

define Landau type quasiparticles the continuity of the mass operator is unavoidably needed). Hence, the CFs are only heuristic construction which displays the multiparticle correlations in the planar charged system at magnetic field presence, and hypothetical CF residual interaction is not derived from the electron repulsion upon any formalism. The interaction is, however, crucial for the 2D topological-type multiparticle correlation creation. The FQHE correlation manifests itself at some specific filling ratios displaying topology constraints in 2D at magnetic field resulting in energy gain over other possible multiparticle states (Wigner crystal insulator, charge density wave, etc). The concept that hypothetical quasiparticles CFs appear or disappear, residually interact or not, due to small shift of density apparently evidences that such quasiparticles actually do not exist but are a convenient representation of topological correlation at specific filling ratios, though with some limits for the usability of the CF picture, especially in higher LLs.

Even though the statement of the authors in lines 30-41 might be vague to some extent, I do not insist for its modification because it is still interesting and worth mentioning.

We thank the Referee for comments on composite fermions and residual CF-CF interactions. Inspired by the Referee, we have slightly modified our manuscript as follows:

In our original manuscript (line 30-41):

“Why might charge-ordered states be expected to supplant FQH and specifically CF states in high LLs? In high ($N>0$) LLs, the more extended electron wave-functions give rise to greater residual interactions²⁴ and may destabilize the FQH states^{21-23, 25}. In GaAs such wave-functions have nodes at particular momenta corresponding to spatial separation between orbitals on the order of the magnetic length and favoring charge ordering with that spacing²⁶. In bilayer graphene, the wave-functions in the $N=2$ LL have no complete nodes, and hence might be expected to support FQH states over charge-ordered states.²⁷ A numerical study which does not rely on the mean-field approximation or otherwise assume the CF picture predicts pronounced single-component FQH states at $1/3$, $2/3$, and $2/5$ (Ref. 27).”

In revised manuscript (line 30-40):

“Why might charge-ordered states be expected to supplant FQH and specifically CF states in high LLs? In high ($N>0$) LLs, the more extended electron wave-functions may destabilize the FQH states²¹⁻²⁵. In GaAs such wave-functions have nodes at particular momenta corresponding to spatial separation between orbitals on the order of the magnetic length and favoring charge ordering with that spacing²⁶. In bilayer graphene, the wave-functions in the $N=2$ LL have no complete nodes, and hence might be expected to support FQH states over charge-ordered states.²⁷ A numerical study which does not rely on the mean-field approximation or otherwise assume the CF picture predicts pronounced single-component FQH states at $1/3$, $2/3$, and $2/5$ in the $N=2$ LL (Ref. 27).”